# The Effect of Mechanical Alteration on Repair Bond Strength of S-PRG-Filler-Based Resin Composite Materials

**DOI:** 10.3390/polym16111488

**Published:** 2024-05-24

**Authors:** Md Sofiqul Islam, Smriti Aryal A C, Shadi El Bahra, Abdullah Jamal Abuhajjeh, Akram Mohammad Al Mofleh, Vivek Padmanabhan, Muhammed Mustahsen Rahman

**Affiliations:** 1Department of Operative Dentistry, RAK College of Dental Sciences, RAK Medical and Health Sciences University, Ras Al-Khaimah P.O. Box 12973, United Arab Emirates; 2Department of Oral and Craniofacial Health Sciences, College of Dental Medicine, University of Sharjah, Sharjah P.O. Box 27272, United Arab Emirates; saryalac@sharjah.ac.ae; 3Department of Prosthodontics, RAK College of Dental Sciences, RAK Medical and Health Sciences University, Ras Al-Khaimah P.O. Box 12973, United Arab Emirates; shadi.elbahra@rakmhsu.ac.ae; 4Department of Pediatric Dentistry, RAK College of Dental Sciences, RAK Medical and Health Sciences University, Ras Al-Khaimah P.O. Box 12973, United Arab Emirates; vivek.padmanabhan@rakmhsu.ac.ae; 5Department of Periodontology, RAK College of Dental Sciences, RAK Medical and Health Sciences University, Ras Al-Khaimah P.O. Box 12973, United Arab Emirates; mustahsen@rakmhsu.ac.ae

**Keywords:** aging of composite, air abrasion, aluminum oxide, diamond bur, glass beads, mechanical alteration, repair bond strength, S-PRG filler, shear bond strength

## Abstract

This study investigates the impact of mechanical alteration on resin composite surfaces and its subsequent effect on repair bond strength. A total of 100 resin composite disks were prepared and were allocated for 24 h or 1 year of artificial aging. Specimens were embedded in epoxy resin, and the composite surfaces were mechanically altered using either diamond burs or air abrasion with aluminum oxide or glass beads. A universal bonding material was applied and a 2 mm circular and 3 mm high repair composite cylinder were prepared using a Teflon mold. Then, the specimens were tested for their shear bond strength, and the de-bonded specimens were observed under a scanning electron microscope to determine the failure pattern. SPSS 26.0 statistical software was used to analyze the data. Two-way ANOVA showed a statistically significant effect of mechanical alteration and aging on the shear bond strength of S-PRG-filler-based resin composite (*p* < 0.05). Surface modification with a fine diamond bur showed a significantly higher bond strength in both 24-h- and 1-year-aged specimens. Surface modification with alumina significantly increased the bond strength of 1-year-aged specimens; however, it was statistically insignificant for 24 h-aged specimens. Mechanical alteration with a fine diamond bur and 50-micron alumina can improve the repair bond strength of the composite.

## 1. Introduction

In dentistry, restoration of defective tooth structure is a routine procedure, and resin composite material is the key ingredient for a successful dental restoration [1]. Historically, dental cements and dental amalgam were the key materials for dental restoration; however, nowadays, resin composite materials have replaced most of them [2]. Since the first innovation, resin-composite-based restorative materials has gone through massive research and development, enabling the production of materials with superior mechanical and esthetic properties. Currently, available resin composite materials can be used for almost all kinds of restoration with very few relative contraindications [3]. The availability of a wide range of color options made it suitable for mimicking tooth color restorative materials. The color blending capability of resin composite materials enables the achievement of a natural tooth-like appearance. The superior esthetic properties of resin composite materials make it a reliable material for both the restoration of tooth defects and the esthetic enhancement of natural dentition [4]. 

The enhanced biocompatibility of resin composite materials made it suitable for restoring deep cavities and restoration of root caries and non-caries cervical lesions. With the advancement of filler technology, the currently available resin composite materials pose high wear resistance properties, which are essential to bear the masticatory load in posterior teeth [5]. The elastic modulus of resin composite materials is similar to that of dentin, which helps its balance between the tensile and compressive strength of the materials. It also prevents fracture of the tooth structure, which was a common issue with metallic restoration like amalgam [6]. 

The development of nano-fill and nano-hybrid filler made the resin composite materials capable of achieving a high polish surface. A highly polished resin composite surface prevents the attachment of bacterial biofilm and external color chromogen and helps in maintaining the esthetic properties of resin composite restoration. These advantages of resin composite materials strengthen the esthetic durability of resin composite restoration [7].

One of the significant advantages of resin composite materials is their capability to restore tooth defects following minimally invasive cavity preparation. It disables the concept of “extension for retention”, which was implemented in amalgam restoration. This might have played a key role in replacing amalgam with resin composite materials. The preservation of non-infected tooth structure and strengthening of the remaining natural tooth structure reduces the necessity of intra-coronal and extra-coronal prosthesis of severely damaged tooth structure [8]. 

Another significant property of resin composite materials is their repair capability. Unlike amalgam, a polymerized resin composite material can be repaired by adding a new layer over the old substrate. This advantage of resin composite material reduces the necessity of replacing the whole restoration, which was a very commonly practiced procedure for amalgam. A repair procedure for resin composite restoration is a common practice to address the issues such as chipping, wear and fracture that took place over time in a resin composite restoration [9]. A repair procedure preserves the natural tooth structure and reduces the risk of compromising the structural stability. It is a cost-effective procedure compared to complete replacement. A repair procedure shortens the clinical procedure time, making it convenient for both the patient and the operator. It also reduces the possibility of developing post-operative sensitivity, which is often challenging in replacing an existing restoration [10]. 

Despite having several advantages, it is quite challenging to achieve a good repair bond strength with aged composite. Resin composite materials undergo several changes over time; the physical and chemical alteration of the restoration surface makes it difficult to bond with a new layer of composite. The resin composite surface can become attached with biofilm and external color chromogen that can contaminate the surface and reduce its binding affinity. The loss of an oxygen-inhibited layer over time reduces the chemical bonding capability of resin composite. A polished resin composite surface reduces the micro-mechanical bonding capability of an old composite with a new one [11]. Considering these factors, it is essential to explore and establish the best repair protocol that can achieve superior bond strength and durable repair. 

Surface pre-reacted glass-ionomer (S-PRG) filler technology was developed and commercialized in the year 2000. The S-PRG filler is a triple-layered glass particle that contains a silicon dioxide coating in the outermost layer beneath it a pre-reacted glass-ionomer phase and a core made of glass. The GIOMER is a restorative resin composite material that uses S-PRG filler technology. This bio-active material is reported to have several advantages over the conventional resin composite such as fluoride release and recharge capability, acid neutralization, promoting mineralization, matrix metalloproteinase (MMPs) inhibition and anti-biofilm properties [12,13]. Having these added benefits, S-PRG-filler-based restorative material is gaining popularity among clinicians, making it a frequently used dental restorative material. Despite having these added benefits, S-PRG-filler-based materials can deteriorate over time. A previous study by Islam et al., 2023, reported the effects of aging of S-PRG-filler-based resin composite material on the optical and physical properties of such materials. In that study, it was reported that the optical/esthetical properties of resin composite changes over time along with the physical properties due to water sorption and solubility of resin composite materials. The aging of S-PRG-filler-based resin composite material can also reduce the surface hardness [5]. Several studies have been conducted on S-PRG-filler-based restorative materials; however, the repair protocol that can achieve superior bond strength and durable repair of S-PRG has not been established yet. Thus, it is essential to experiment with different parameters that might have an influence on the repair bond strength of S-PRG-filler-based resin composite material to establish the best protocol for repairing the restoration. 

In this in vitro study, several mechanical surface alterations were employed to evaluate their impact on the repair bond strength to S-PRG-filler-based resin composite materials. The objective was to evaluate and compare the repair shear bond strength (SBS) of 24-h and 1-year-aged S-PRG-filler-based resin composite material. The first null hypothesis tested in this research was that there is no difference in repair SBS to S-PRG-filler-based resin composite materials following different mechanical surface alterations. The second null hypothesis was that artificial aging would not negatively affect the repair SBS to S-PRG-filler-based resin composite materials.

## 2. Materials and Methods

**Study design:** This in vitro study was conducted using S-PRG-based resin composite materials. This study protocol was reviewed and approved by the research and ethical committee of Ras Al Khaimah College of Dental Sciences with the approval number RAKCODS-REC-07-2023/24-UG. The methodological illustration of the study is shown in Figure 1.

**Preparation of resin composite substrate**: A total of 100 disks were prepared using S-PRG-filler-based resin composite materials (Beautifil II LS, shade A3.5 Shofu Dental Inc., Higashiyama-ku, Kyoto, Japan). Seven-millimeter-wide and three-millimeter-thick disks were prepared by polymerizing the resin composite materials in a Teflon mold. The resin composite materials were packed in the Teflon mold using composite placing instruments and covered and pressed using a glass slab to obtain uniform thickness and a smooth surface. The disks were polymerized from both sides using a light curing device (Paradigm™ Deep Cure LED Curing Light, 3 M Oral Care, St. Paul, MN, USA) for 40 s.

**Aging of resin composite disk:** The polymerized resin composite disks were randomly divided into 2 groups. Fifty resin composite disks were stored in distilled water for 24 h at a temperature of 37 °C. The remaining 50 specimens were allocated for artificial aging using a thermos-cycling device (Thermocycler THE-1100, SDMECHATRONIK GMBH, Feldkirchen-Westerham, Munich, Germany). During each cycle, the specimens were immersed in water at 5 °C for 30 s and then taken out of water and rested for 5 s, followed by in 55 °C water for 30 s. A total of 10,000 thermal cycles were performed to age the specimens to 1 year old [5]. 

**Embedding of resin composite disks:** After 24 h and 1 year of aging, the specimens were embedded in an epoxy resin. The specimens were placed upside down in a 25 mm circular and 15 mm-high mold. The resin and hardener were mixed following the manufacturer’s instructions, poured into the mold, and allowed harden completely over 48 h. Then, the specimens were removed from the mold and the surfaces were cleaned and gently polished to remove any debris attached to the resin composite surfaces.

**Surface modification:** The 24 h- and 1-year-aged specimen were randomly divided into 5 groups (*n* = 10 + 10). Then, the specimen surfaces were mechanically modified using one of the protocols mentioned in Table 1 except for the control group. In group 2, the surface was altered using a uni-directional stroke with a fine diamond bur (EX-21F, MANI. INC, Takanezawa, Shioya, Tochigi, Japan) with a particle size between 53 and 63 µ, and group 5 specimens were altered using an extra fine diamond bur (EX-21EF, MANI. INC, Tochigi, Japan) with a particle size between 20 and 30 µ connected to a high-speed dental hand piece (Pana Air FX, NSK, Kanuma-Shi, Tochigi, Japan). The group 3 specimen’s surfaces were altered with 50 µ aluminum oxide (EDELKORUND, Eisenbacher, Wörth am Main, Germany), and the group 4 specimens were altered with 90 µ glass beads (DANVILLE, Carlsbad, CA, USA) using air-borne particle abrasion (APA) from a constant distance and uniform pressure (6 bar).

**Repair of the substrate:** After surface treatment, the specimen surfaces were cleaned using 37% phosphoric acid for 20 s. After rinsing and air drying, an all-in-one universal adhesive (Beauti Bond Xtreme, Shofu Dental Inc., Japan) was applied using a micro-brush. The bonding was agitated well on the surface using a scrubbing motion. After that, gentle air drying was applied for 5 s. The bonding layer was polymerized using a light cure for 20 s. A 3 mm-thick Teflon mold with a 2 mm circular punched hole was placed in the center of the bonded substrate and stabilized by holding it with tweezers. Repair resin composite material (Beautifil Flow Plus F00, Shade A1, Shofu Dental Inc., Japan) was injected into the punched hole to fill it. Then, it was polymerized using light curing for 20 s. After that, the mold was removed, and the specimen received an additional light cure for 20 s. 

**Shear bond strength testing**: After repair, the specimens were allocated for shear bond strength testing. A blinded operator (not aware of specific surface treatment applied on individual specimens) performed the shear bond strength. The specimens were mounded in the shear test device at the same level and parallel to the direction of force. A semi-circular metal attachment unit that deploys the shear force was placed close to the repaired composite cylinder. The shear forces were applied at the bonded interface until failure at a crosshead speed of 1.0 mm/min. The shear force required to de-bond the specimens was recorded in Kgf. The shear bond strength value was calculated by dividing the required force by the surface area and was then converted to a megapascal (MPa) value.

**Failure mode analysis:** After shear bond testing, the de-bonded specimens were allocated for failure mode analysis. Each specimen was observed under a stereo microscope (Olympus BX53, Shinjuku, Tokyo, Japan) at 40× magnification. Then, the specimens were dehydrated in a desiccator for 24 h. The specimen surface was then coated with Gold/Palladium (80% and 20%), 57 mm Ø × 0.1 mm-thick in a vacuum chamber (10–2 m bar pressure) with the presence of Argon gas and an 18 mA plasma current for 120 s in a gold spatter device (Quorum Technology Mini Sputter Coater, SC7620, East Sussex, UK). The specimens were then observed using a scanning electron microscope (SEM) (Tescan VEGA XM variable pressure SEM, Kohoutovice, Czech Republic) at accelerating voltage: Max. 30 kV and at 75× and 500× magnification. 

**Surface morphology observation:** An additional 5 specimens (1 year aged) were used for the surface morphology observation. Each specimen surface was altered using one of the mechanical alteration methods. Then, the specimens were dehydrated in a desiccator for 24 h. The specimen surface was then coated with a gold spatter. The specimens were then observed using a scanning electron microscope at 1000× magnification to evaluate the surface morphology.

**Data analysis:** The shear bond strength date was input and analyzed using statistical software (SPSS.24.0, IBM, Armonk, NY, USA). A descriptive analysis was performed to evaluate the data distribution. Two-way ANOVA was performed to evaluate the effect of surface modification and aging on SBS. One-way ANOVA was performed to for both the 24 h- and 1-year-aged groups to evaluate the statistical significance among the tested groups. The Tukey post hoc test was performed for the multiple comparisons among the tested groups at a 95% confidence level.

## 3. Results

### 3.1. Shear Bond Strength

Two-way ANOVA showed a statistically significant effect of surface modification (*p* = 0.001) and aging time (*p* = 0.001) on the SBS of the resin composite materials and an interaction between the surface modification used and the aging time (*p* = 0.001). One-way ANOVA showed statistical significance between the 24 h-aged groups (*p* = 0.001). Surface alteration with a fine diamond bur showed a statistically significant increase in SBS compared to the control group using the Tukey post hoc test (*p* = 0.002). The use of 50-micron alumina and 90-micron glass beads showed an elevated SBS value compared to the control group; however, it was statistically insignificant (*p* = 0.493 and *p* = 0.185). The use of an extra-fine diamond bur neither increased nor decreased the SBS compared to the control group (*p* = 1.0). The mean SBS and standard deviation (SD) of each experimental group are shown in Figure 2. 

In the case of 1-year-aged groups, one-way ANOVA showed a statistical significance between the tested groups (*p* = 0.001). The Tukey post hoc test revealed that surface alteration with a fine diamond bur significantly increased the SBS compared to the control group (*p* = 0.040). Using 50-micron alumina and 90-micron glass beads produced an elevated SBS value compared to the control group; however, this was statistically insignificant (*p* = 0.122 and *p* = 0.171). The use of an extra-fine diamond bur neither increased nor decreased the SBS compared to the control group (*p* = 0.999). The mean SBS and SD of each experimental group are shown in Figure 3. 

Multiple comparisons between 24-h- and 1-year-aged groups showed that the SBS significantly reduced in the 1-year-aged groups compared to its corresponding 24-h-aged groups (*p* < 0.05). The data comparison based on surface alteration aging time is shown in Table 2. The multiple comparisons of all ten groups are shown in Figure 4.

### 3.2. Failure Mode

Based on microscopic observation, the failure pattern was subdivided into four categories: (a) cohesive failure within substrate, (b) adhesive failure, (c) mixed failure, and (d) cohesive failure within repair composite. The graphical explanation of the failure mode is shown in Figure 5. In the 24 h-aged control group and alumina teared group, the majority of the specimens showed an adhesive failure to the substrate. In the fine diamond bur-treated group, 50% of specimens showed cohesive failure within the substrate, and the remaining 50% showed adhesive failure to the substrate. The detailed failure pattern of 24-h-aged groups is shown in Figure 6. 

In the case of 1-year-aged groups, cohesive failure within the repair composite was predominant in most of the groups except the control group, where the majority of the specimens showed mixed failure. In the extra-fine diamond bur-treated group, 50% of the specimens showed cohesive failure within the repair composite, and the remaining 50% showed mixed failure. None of the specimens showed cohesive failure within the substrate in 1-year-aged groups. Details on the failure pattern of the 1-year-aged groups are shown in Figure 7. The representative SEM image of each failure pattern is shown in Figure 8.

### 3.3. Surface Morphology

Surface alteration with a fine diamond bur showed an irregular and deep scratched surface under SEM observation. In the case of an extra-fine diamond bur being utilized, the irregularities and scratched were less prominent. Surface alteration using APA with aluminum oxide and glass beads showed a uniform surface erosion pattern under SEM observation. The representative SEM image of each group after surface alteration is shown in Figure 9.

## 4. Discussion

The control groups, non-aged and aged, did not receive any individual mechanical surface treatment. The repair process was carried out in successive steps: cleaning the bonding surface with 37% phosphoric acid etchant [14], application of adhesive bonding agent, and repairing with resin composite (to simulate the regularly followed clinical procedure). Accordingly, the control groups served as a reference, i.e., negative control, for all experimental mechanically treated groups.

The shear test is one of the most widely used macro-bonding test methods for evaluation of bond strength due to its simplicity and ease of implementation [15]. In this study, a specially designed jig was utilized. This jig was equipped with a knife-edge chisel to exert the shear force. This kind of test setup could have a critical impact on stress distribution/concentration and, consequently, might be a source of inconsistency in shear bond measurements [16,17,18]. Moreover, SBS calculated by dividing the peak failure load by the cross-sectional area might underestimate the true interfacial bond strength [19]. 

Furthermore, it has been revealed that cohesive failure of the substrate or composite resin material, or mixed failure of both, represents an intrinsic disadvantage of shear bond testing due to the high variance in stress distribution. This unevenness in stress concentration can initiate a crack that propagates within the substrate/resin composite/bonding interface and cause monolithic fractures [19,20]. Nevertheless, several studies pertinent to the repair bond strength to resin composites have employed the SBS test [21,22,23,24,25].

In order to minimize the risk of bias, certain measures were taken. Blinding the operator who conducted the shear mechanical test was an integral part of this study. Additionally, obtaining standard specimens was taken into consideration by means of preparing the specimens by one operator while another operator performed the standard specimen selection [26]. 

Regarding the polymerization process of resin composites, it is determined that the number of unsaturated double bonds decreases with the lapse of time, i.e., aging [27,28]. The presence of these reactive double bonds is essential for establishing a reliable and durable adhesion to the existing “old” resin composite substrates [29,30]. Some artificial aging modalities like thermos cycling and water storage were utilized in laboratory studies to simulate the hydrolytic degradation of restorative resin composites in the clinical setting [31,32]. In the oral cavity, temperature changes are related to the body temperature and temperature fluctuating due to the hot or cold food and beverages consumed by the individual. Based on that, it is well established that the protocol to age the dental restorative materials should be within the range of 5 °C and 55 °C. In order to mimic the conditions of the oral environment, it is recommended to use 5 °C and 55 °C for the ageing of restorative dental materials, although it does not represent the glass transition temperature of the particular composite material. It has been reported that 5000 thermal cycles were most frequently employed for the thermos-cycling procedure [33]. Therefore, for better simulation of the aging process occurring in the oral cavity, half of the investigated resin composite specimens in the current study were submitted to 10,000 thermal cycles, which correspond to 12 months of clinical service [34].

To effectively enhance the reliability and durability of the repair bond strength, mechanical treatment of the substrate surface prior to chemical conditioning with coupling agents (silane) and/or adhesive resins (conventional/universal) has been highly recommended [35,36,37]. The purpose of the mechanical treatment is to enlarge the surface area, improve the surface wettability, and obtain enhanced surface roughness [38]. Consequently, this would probably promote strong mechanical interlocking with the surface irregularities [39,40]. Mechanical treatment was achieved by various means including using diamond burs with different levels of coarseness, or via APA using different types of abrasive mediums and protocols.

The first null hypothesis was rejected, as the repair SBS to S-PRG-filler-based resin composite substrate was influenced by the applied mechanical surface treatment. Regarding the non-aged specimens, bonding surfaces treated via blasting with either 50 µ alumina particles or 90 µ glass beads improved the repair SBS mean values in comparison to the values obtained by the non-treated surfaces. However, this increase was not statistically significant. Only bonding surfaces treated with a fine diamond bur demonstrated the highest, statistically significant, repair SBS mean values in comparison to non-treated surfaces. Interestingly, bonding surfaces treated with an extra-fine diamond bur, when compared to non-treated surfaces, exhibited lower but statistically insignificant repair SBS mean values. This could indicate that not every mechanical surface treatment could represent a reliable modality. An SEM images in Figure 9 displays typical macro-retentive irregular features of a substrate surface treated with a fine diamond bur. An SEM image in Figure 9 shows less irregular, typical micro-retentive features of a substrate surface blasted with 50 µ alumina particles. 

The results of the present study are in agreement with the results obtained by Wendler et al., 2016, who found significant differences in the repair bond strength mean values among surfaces treated with diamond burs of varying coarseness [14]. They suggested that the larger the grain size, the rougher the surface, and the higher the repair bond strength. Moreover, they disclosed that the application of bonding adhesive to previously roughened surfaces with a red- or blue-coded diamond bur led to significantly higher bond strengths than those surfaces subjected to APA with 35 µ alumina particles. 

Our results are inconsistent with the findings of Da Costa et al., 2012, who stated that the variations in the grain size of diamond burs did not affect the repair bond strength [38]. Moreover, they revealed that APA with 50 µ alumina particles showed the highest repair bond strength. Furthermore, Dall’oca et al., 2008, reported that repair surfaces treated solely with 37% phosphoric acid demonstrated the highest statistically significant micro-tensile bond strength [41]. Additionally, APA with 50 µ alumina particles revealed a higher statistically significant micro-tensile bond strength than mechanical treatment with a medium-grit bur. In this regard, it is worth noting that the dental literature reports many controversial and conflicting findings. This may be attributed to several crucial factors such as, but not limited to, the composition and structure of the investigated resin composites and bonding adhesives, the variations in the implemented repair protocols, and the employed experimental test settings [42,43,44,45,46].

To the best of the authors’ knowledge, APA with glass beads has not yet been used as a viable option in repair procedures for direct resin composites. Mehari et al., 2020, investigated the impact of the blasting medium type on the SBS of resin cement to different generations of dental zirconia [47]. They concluded that APA with glass beads resulted in significantly lower SBS of the resin cement to all studied zirconia generations compared to APA with alumina. On the contrary, Nobuaki et al., 2015, reported that blasting with glass beads resulted in significantly superior SBS between resin cements and CAD/CAM resin composites when compared to blasting with alumina particles [48]. Moreover, resin composite surfaces abraded with glass beads demonstrated less damage compared to that caused by alumina particles. In the present study, obtained SEM images (Figure 9) are in accordance with the SEM findings illustrated in Nobuaki’s study. APA with alumina particles created a highly irregular surface texture compared to the surface texture generated by glass beads [41]. However, regardless of the aging condition, substrates blasted either by alumina particles or glass beads exhibited comparable SBS mean values, which did not significantly differ from the control group. A potential reasonable explanation for these findings could be that resin composite substrates contained sufficient amounts of reactive double bonds, even after aging, to establish a reliable and durable bond between the “old” and the “new” resin composite [29,30]. 

It is well documented that artificial aging, i.e., thermos cycling, initiates both hygroscopic and hydrolytic changes in restorative resin composites [31,49]. Additionally, as earlier mentioned, the content of unsaturated double bonds declines with time [27,28]. Thus, the bonding behavior to aged substrates was anticipated, and the significant reduction in SBS mean values was not surprising (Table 2). Therefore, the second null hypothesis was rejected. Bonding surfaces treated with a fine diamond bur exhibited the highest repair SBS mean values. However, they did not significantly differ from repair SBS mean values demonstrated by the surfaces blasted with 50 µ alumina particles. This indicates that both mechanical treatments can effectively contribute to establishing a reliable repair BS to aged resin composite restorations. No significant differences in repair SBS mean values were detected among non-treated surfaces, surfaces treated with 90 µ glass beads, and surfaces treated with an extra-fine diamond bur. Although mechanical treatment with glass beads performed well on non-aged substrates, the performance on aged substrates was not satisfactory. This likely suggests that repair BS to surfaces blasted with glass beads relied solely on chemical bonds rather than micromechanical retention. This interpretation can also be extrapolated to the repair BS to surfaces treated with an extra-fine diamond bur. SEM images in our study (Figure 9) reveal smooth surface texture associated with the aforementioned mechanical treatments: glass beads and extra-fine diamond burring. On the contrary, APA treatment with 50 µ alumina particles or roughening the bonding surface with a fine diamond bur created a different surface roughness pattern (Figure 9), which might improve the wettability of the low viscosity universal adhesive and could enhance the penetration and interlocking of the injectable repair resin composites into surface irregularities, thus obtaining higher SBS mean values [48]. 

De-bonding of the specimen during bond strength testing occurs due to stress concentration in the weakest portion of the joint, which acts as a point of crack initiation. With increased stress, it propagates, finally causing fracture or de-bonding [50]. The variation in failure pattern in 24 h-aged and 1 year-aged observed in our study might be associated with the bond strength and surface characteristics [51]. The difference in surface properties of a 24 h-aged substrate with that of a 1 year-aged changed the location of crack initiation.

**Study limitations:** In this study, the used universal adhesive bonding agent and S-PRG-filler-based resin composite materials were from the same manufacturer. Materials from other manufacturers and different types of filler-based resin composite material should be investigated, since, in most clinical scenarios, the composition/type of resin composite is unknown. Moreover, additional experimental settings could be investigated, such as glass beads or alumina particle of different sizes, the employed blasting pressure, and higher coarseness of diamond burs. Furthermore, aging of bonding interfaces after conducting the repair procedure was not performed.

## 5. Conclusions

Aging of S-PRG-filler-based resin composite substrate can reduce the bonding capability during the repair procedure. Mechanical alteration of S-PRG-filler-based resin composite substrate using fine diamond bur or air-born particle abrasion using 50 µ aluminum oxide can improve the bonding strength during the repair procedure. Based on the results of this study, it can be concluded that:The bonding (repair) capability of S-PRG-filler-based resin composite materials reduces over time;In order to obtain optimal bond strength, it is essential to mechanically modify the surface of aged S-PRG-filler-based resin composite substrate;Surface alteration with a fine diamond bur can enhance the repair bond strength of S-PRG-filler-based resin composite materials;Surface alteration with APA using 50 µ aluminum oxide can enhance the repair bond strength of aged S-PRG-filler-based resin composite materials;Surface alteration with APA using 90 µ glass beads can improve the repair bond strength of 24 h-aged S-PRG-filler-based resin composite materials, but it negatively affects the repair bond strength of a 1-year-old substrate;Surface alteration with a super-fine diamond bur has no significant effects on the repair bond strength of S-PRG-filler-based resin composite materials.

## Figures and Tables

**Figure 1 polymers-16-01488-f001:**
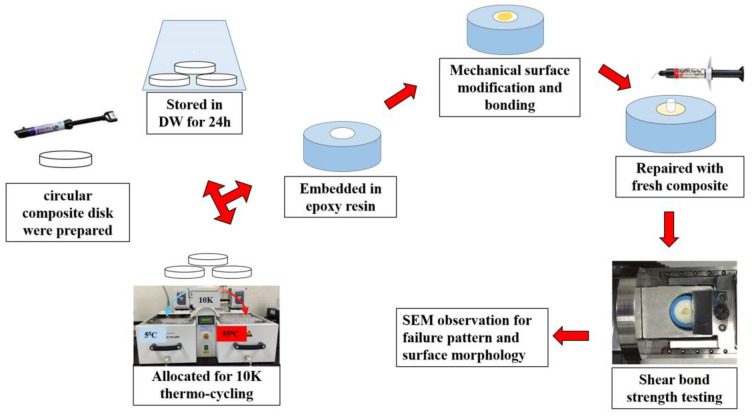
Experiment design.

**Figure 2 polymers-16-01488-f002:**
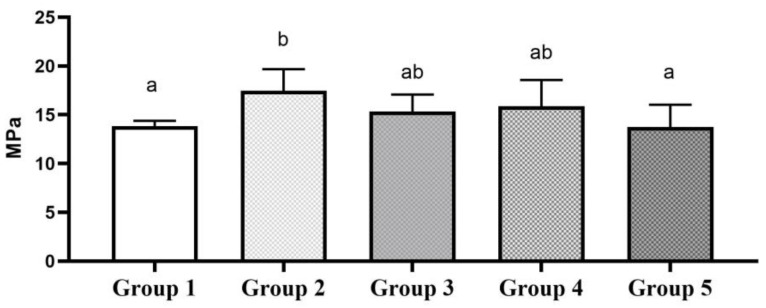
SBS of 24 h-aged groups. Bar graphs identified with same alphabet are statistically insignificant (*p* > 0.05).

**Figure 3 polymers-16-01488-f003:**
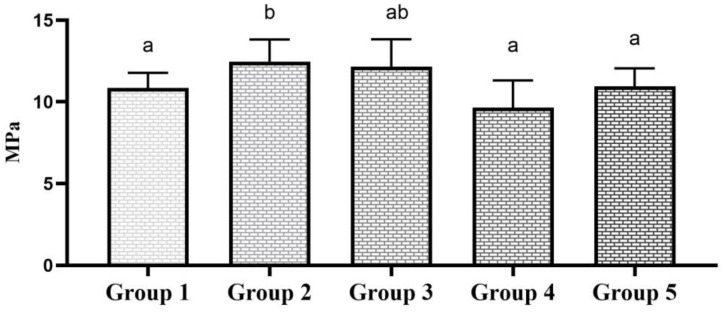
SBS of 1 year-aged groups. Bar graphs identified with same alphabet are statistically insignificant (*p* > 0.05).

**Figure 4 polymers-16-01488-f004:**
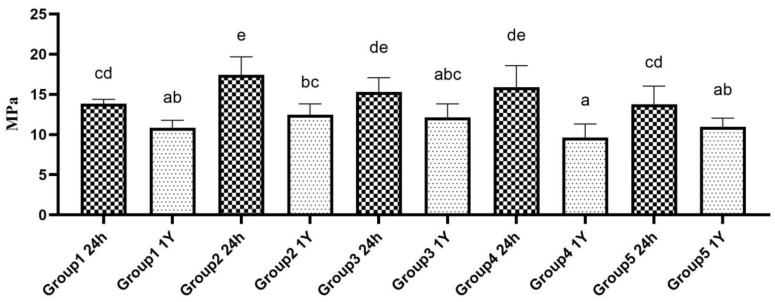
Multiple comparison on SBS of both 24 h-aged and 1 year-aged groups. Bar graphs identified with same alphabet are statistically insignificant (*p* > 0.05).

**Figure 5 polymers-16-01488-f005:**
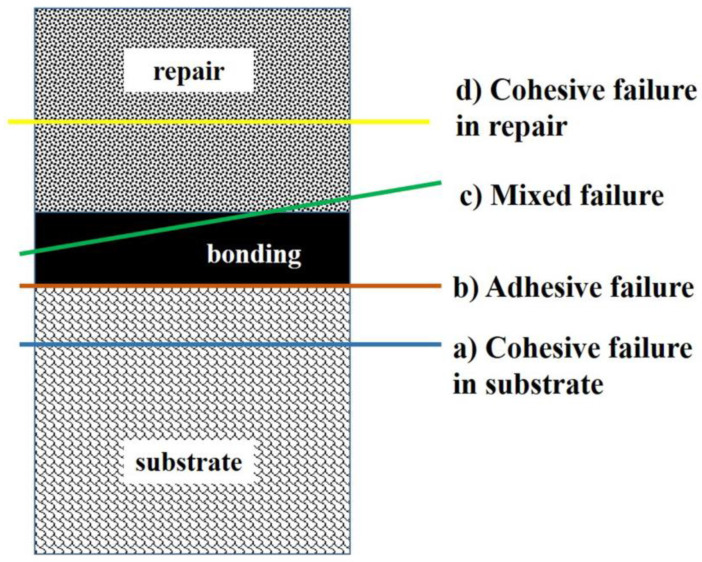
Schematic diagram of failure pattern.

**Figure 6 polymers-16-01488-f006:**
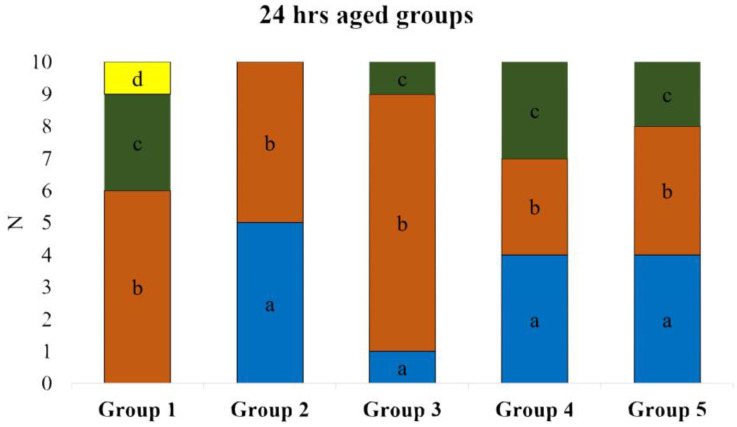
De-bond pattern of 24 h-aged groups. (a) Cohesive failure within the substrate; (b) adhesive failure; (c) mixed failure; (d) cohesive failure within repair composite.

**Figure 7 polymers-16-01488-f007:**
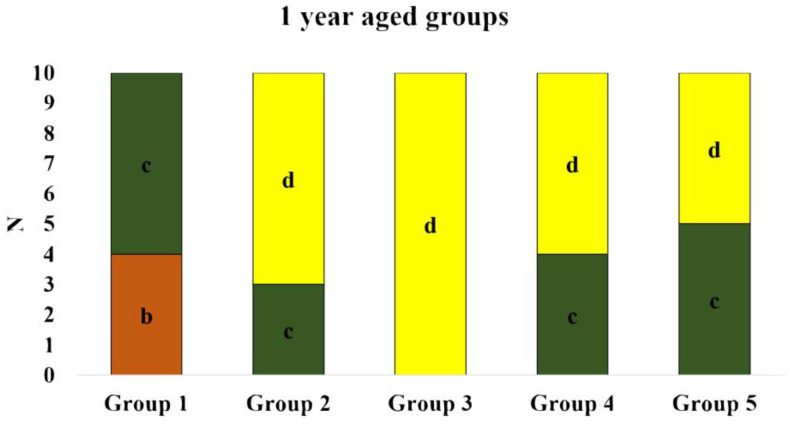
De-bond pattern of 1 year-aged groups. (a) Cohesive failure within the substrate; (b) adhesive failure; (c) mixed failure; (d) cohesive failure within repair composite.

**Figure 8 polymers-16-01488-f008:**
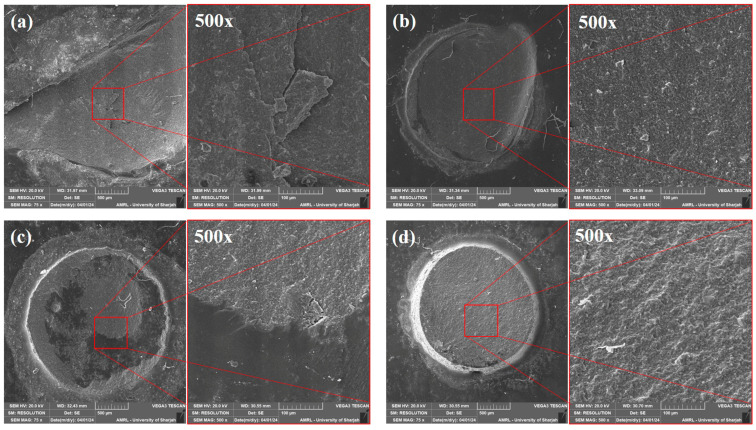
Representative SEM images of each failure pattern at 75× and 500× magnification. (**a**) cohesive failure within the substrate; (**b**) adhesive failure; (**c**) mixed failure; (**d**) cohesive failure within repair composite.

**Figure 9 polymers-16-01488-f009:**
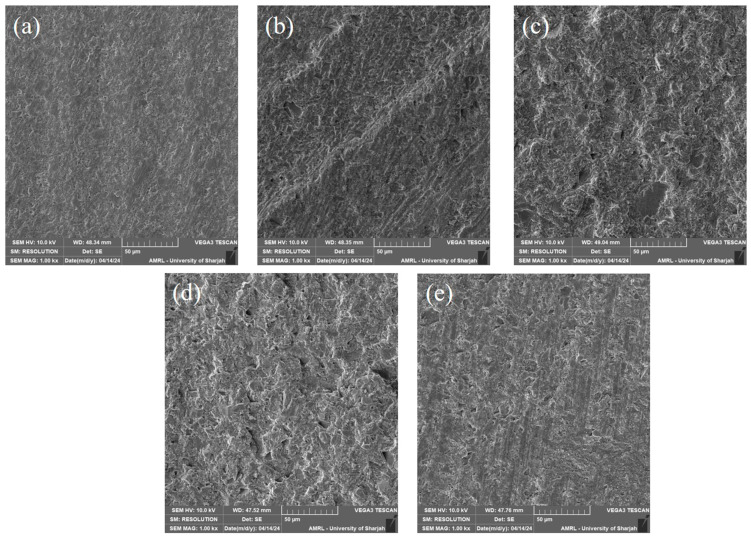
SEM images showing the surface morphology after mechanical alteration at 1000× magnification. (**a**) Control; (**b**) fine diamond bur; (**c**) APA with aluminum; (**d**) APA with glass beads; (**e**) extra-fine diamond bur.

**Table 1 polymers-16-01488-t001:** Group distribution and surface modification protocol.

Group	Mechanical Surface Alteration Protocol
1	Phosphoric acid etching
2	Fine diamond bur + phosphoric acid etching
3	APA (alumina 50 micron) + phosphoric acid etching
4	APA (glass beads 90 micron) + phosphoric acid etching
5	Super-fine diamond bur + phosphoric acid etching

**Table 2 polymers-16-01488-t002:** Shear bond strength with means and standard deviations in MPa for all tested groups.

	Group 1	Group 2	Group 3	Group 4	Group 5	*p*
24 h-aged substrate	13.85 ± 0.54 ^A,a^	17.46 ± 2.22 ^A,b^	15.33 ± 1.75 ^A,ab^	15.89 ± 2.69 ^A,ab^	13.76 ± 2.27 ^A,a^	0.001
1 year-aged substrate	10.86 ± 0.92 ^B,a^	12.47 ± 1.34 ^B,b^	12.16 ± 1.66 ^B,ab^	9.65 ± 1.66 ^B,a^	10.96 ± 1.1 ^B,a^	0.001
*p*	0.008	0.001	0.004	0.001	0.018	

Significance determined at α = 0.05. Different superscript uppercase letters indicate significant differences within each column (*p* ≤ 0.05). Different superscript lowercase letters indicate significant differences within each row (*p* ≤ 0.05).

## Data Availability

The raw data supporting the conclusions of this article will be made available by the corresponding author upon request.

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
