# Peer review of "The Effect of Mechanical Alteration on Repair Bond Strength of S-PRG-Filler-Based Resin Composite Materials"

_polymers, 2024, doi:10.3390/polym16111488_

Round 1

Reviewer 1 Report

Comments and Suggestions for Authors

Comments

My comments for the manuscript entitled, ‘The Effect of Mechanical Alteration on Repair Bond Strength of S-PRG Filler-based Resin Composite Materials’ are given below,

1.       The manuscript does not deal either about the polymer or polymer processing.

2.       The curing process of S-PRG is not explained properly and not understandable in the present form.

3.       Mechanical properties of the prepared samples are not given. Include the figures obtained for mechanical properties.

4.       SEM images of the failure pattern shows only the top-view image, but, failure pattern will be clearer with the cross-sectional images. Include the cross-sectional images of both failure pattern and non-failure pattern.

5.       It is mentioned that supplementary materials are provided. But there are no supplementary materials found. Include it.

6.       Conclusion part is too short. Elaborate the conclusion part with the results obtained highlighting with the obtained values.

Author Response

The authors really appreciate the in-depth review and the valuable comments of the reviewer. The comments were significantly helpful in improving the manuscript.

Comment: The manuscript does not deal either about the polymer or polymer processing.

Reply: Thank you for the comment. The author agrees with the reviewer’s comment. The objective of the manuscript was to evaluate the effect of the mechanical surface alteration 24h and 1-year aged S-PRG filler-based resin composite material (a methacrylate-based material used for tooth restoration) on repair bond strength.

Comment: The curing process of S-PRG is not explained properly and not understandable in the present form.

Reply: Thank you for the comment. S-PRG filler-based resin composite material is a routinely used dental restorative material. The curing process of this material is very simple. The curing/ polymerization of this material is based on a photochemical reaction. As mentioned in the manuscript after application of the material, it needs to be irradiated with visible light that initiates the polymerization process.

Comment: Mechanical properties of the prepared samples are not given. Include the figures obtained for mechanical properties.

Reply: Thank you for the comment. In this study, we did not evaluate the mechanical properties of S-PRG filler-based resin composite materials. This study investigated the mechanical surface alteration effect on repair bond strength. The optical and physical properties of this material have been reported in our previous article published in Polymer Journal in 2023. The article has been cited in the manuscript (reference number 5).

Comment: SEM images of the failure pattern shows only the top-view image, but, failure pattern will be clearer with the cross-sectional images. Include the cross-sectional images of both failure pattern and non-failure pattern.

Reply: Thank you for pointing out an important aspect. The reviewer is absolutely right. This is a very common technique applied in evaluating the failure pattern of de-bonded specimens after a micro-tensile bond strength test where the specimen needs to be sectioned to obtain a beam/ hourglass shape. However, in our study due to the different diameters of the substrate (old composite) and repair (new composite), it was not possible to obtain the cross-sectional view of de-bonded specimens. Moreover, since the bonding interface between the substrate and the repair resin composite materials was the center of study attention, the focus was on capturing SEM images of this particular area.  

Comment: It is mentioned that supplementary materials are provided. But there are no supplementary materials found. Include it.

Reply: Thank you for pointing out the incomplete/ missing statement. All the data obtained in this study has been presented as tables and bar graphs within the manuscript. This section has been revised.

Comment: Conclusion part is too short. Elaborate the conclusion part with the results obtained highlighting with the obtained values.

Reply: Thank you for the comment. The conclusion section has been revised.

Reviewer 2 Report

Comments and Suggestions for Authors

Use the full name form of the abbreviations where you use them for the first time.

The novelty of the work must be highlighted in the last paragraph of the introduction.

In the materials and methods section, you mentioned the preparation method, aging while the relevant figures and tables are in the results section. Revise this.

The experiment design cannot be in the results section.

The importance of this type of ageing must be described in the manuscript.

What standard test procedure did you employ to carry out the ageing process?

How did you select the temperatures?

Did you consider the glass transition temperature of the resin?

What about the water uptake in the samples? Did you measure that?

There is nothing about the degradation mechanisms in the samples.

Add scale bar to the SEM images.

The discussion section is more like a literature review.

Use bullets in the conclusion section to highlight the main achievements of the work.

 Conclusion section is too short.

The introduction needs to be improved. There are many recently published papers regarding ageing and degradation in composite materials such as the below ones. 

Multi-scale modelling and life prediction of aged composite materials in salt water

Hygrothermal aging of particle-filled epoxy-based composites

Author Response

The authors really appreciate the in-depth review and the valuable comments of the reviewer. The comments were significantly helpful in improving the manuscript.

Comment: Use the full name form of the abbreviations where you use them for the first time.

Reply: Thank you for the comment. The abbreviations have been revised.

Comment: The novelty of the work must be highlighted in the last paragraph of the introduction.

Reply: Thank you for the suggestion. The novelty of the work has been added in the introduction section (lines 102-107).

Comment: In the materials and methods section, you mentioned the preparation method, aging while the relevant figures and tables are in the results section. Revise this.

Reply: Thank you for the comment. This position of table and figure has been revised.

Comment: The experiment design cannot be in the results section.

Reply: Thank you for the comment. The illustration of the experiment design has been changed as per the suggestion.

Comment: The importance of this type of ageing must be described in the manuscript.

Reply: Thank you for the comment. The suggested section has been added (line number 313-319).

Comment: What standard test procedure did you employ to carry out the ageing process?

Reply: Thank you for the comment. The aging protocol used in this study has been implemented in several previously published articles. A citation has been added to refer to the particular article.

Comment: How did you select the temperatures?

Reply: Thank you for the comment. The temperature was selected following a previously published article. A citation has been added to refer to the particular article. In the oral cavity, the temperature changes are related to the body temperature and temperature fluctuating due to the hot or cold food and beverages consumed by humans. Based on that it is well established that the protocol to age the dental restorative materials should be within the range of 50C and 550C. 

Comment: Did you consider the glass transition temperature of the resin?

Reply: Thank you for the comment. It is a valid point to be considered while performing the aging of any composite materials. However, as we explained in the reply to the previous comment, in order to mimic the conditions of the oral environment, it is recommended to use 50C and 550C for the aging of restorative dental materials although they do not represent the glass transition temperature of the particular composite material. Furthermore, the aging process of the substrate material was conducted before proceeding with the repair process and without applying any load on the substrate specimens. Lastly, the subsequent repair procedure and shear bond strength (SBS) test were carried out for all substrate specimens at room temperature, which, most probably, will have a negligible impact on the glass transition temperature of the investigated resin materials.

Comment: What about the water uptake in the samples? Did you measure that?

Reply: Thank you for the comment. We have not measured the water uptake in the samples in this study. However, the water uptake of this material has been reported in our previous article published in Polymer Journal in 2023. The article has been cited in the manuscript (reference number 5).

Comment: There is nothing about the degradation mechanisms in the samples.

Reply: Thank you for the comment. The degradation mechanism of the sample has been mentioned in lines (313-319). Again, in this study we tried to mimic the repair process under clinical setting conditions and the bond strength to fresh (non-aged) and old (aged) resin composite restorations, therefore, the degradation mechanism (solubility/constituents leach out) and the related tests were out of the scope of the study.

Comment: Add scale bar to the SEM images.

Reply: Thank you for the comment. The default scale bars show the actual scale and the test condition is already shown in each SEM image.

Comment: The discussion section is more like a literature review.

Reply: Thank you for the comment. In the discussion section, we attempted to give a rational explanation of the methodology used in this study in view of previously published articles on similar topics or setups. The second section of the discussion was intended to give a possible explanation of the data obtained in this study and to link them with previously published articles where the researcher showed similar or contrary data. The logical explanation/ mechanism behind obtaining the result was discussed in detail in this section. However, the discussion section has been carefully reviewed and revised as per the suggestion of the reviewer.

Comment: Use bullets in the conclusion section to highlight the main achievements of the work.

Reply: Thank you for the comment. The conclusion section has been revised.

Comment: Conclusion section is too short.

Reply: Thank you for the comment. The conclusion section has been revised.

Comment: The introduction needs to be improved. There are many recently published papers regarding ageing and degradation in composite materials such as the below ones. 

Multi-scale modelling and life prediction of aged composite materials in salt water

Hygrothermal aging of particle-filled epoxy-based composite

Reply: Thank you for referring to two interesting articles. We have gone through the articles. Both of the articles are very informative. It explains the mechanical properties and degradation process of epoxy-based composite material. However, the S-PRG filler-based resin composite materials used for tooth restoration are different from the composite materials used in those studies. Unfortunately, the explanation given in those articles might not be applicable to the composite materials investigated in our study. 

Round 2

Reviewer 1 Report

Comments and Suggestions for Authors

The authors responded to my comments. I accept the current version of the publication

Author Response

On behalf of all the authors, I would like to thank the reviewer for the kind acceptance of the manuscript. 

Reviewer 2 Report

Comments and Suggestions for Authors

The introduction has not been improved at all.

More research regarding the aging process needs to be included in the introduction.

Author Response

The authors appreciate the reviewer's comments on the revised version of the manuscript. As per the reviewer's recommendation, we have added a paragraph about the aging effect of S-PRG filler-based resin composite materials in the introduction section (lines 100-108). The authors want to admire the in-depth review and valuable comments of the reviewer that played a significant role in improving the manuscript.